

# Measurement matters: higher waist-to-hip ratio but not body mass index is associated with deficits in executive functions and episodic memory

Andree Hartanto[1] and Jose C. Yong[1,2]

[1] School of Social Sciences, Singapore Management University, Singapore, Singapore
[2] National University of Singapore, Singapore, Singapore

## ABSTRACT

**Background:** The current study aimed to reconcile the inconsistent findings between obesity, executive functions, and episodic memory by addressing major limitations of previous studies, including overreliance on body mass index (BMI), small sample sizes, and failure to control for confounds.
**Methods:** Participants consisted of 3,712 midlife adults from the Cognitive Project of the National Survey of Midlife Development. Executive functions and episodic memory were measured by a battery of cognitive function tests.
**Results:** We found that higher waist-to-hip ratio was associated with deficits in both executive functions and episodic memory, above and beyond the influence of demographics, comorbid health issues, health behaviors, personality traits, and self-perceived obesity. However, higher BMI was not associated with deficits in executive functions and episodic memory. More importantly, these differential associations were robust and stable across adulthood.
**Discussion:** Our findings confirm the association between obesity and episodic memory while highlighting the need for better measures of obesity when examining its associations with individual differences in cognitive functions.

## INTRODUCTION

High levels of adiposity, manifested as being overweight to obese, is a significant and growing challenge to public health (*Clark et al., 2016*). As a recognized risk factor for a wide range of chronic diseases such as metabolic syndrome, diabetes, hypertension, cardiovascular disease, stroke, and certain types of cancer (*Kopelman, 2000*), obesity carries a heavy economic burden for governments and healthcare providers across the world (*Avenell et al., 2004*). Concerns over the dangers of obesity are further heightened by its increasing prevalence in both developed and developing countries, with approximately 30% of the world's population being either obese or overweight (*Ng et al., 2014*).

While obesity undoubtedly correlates with various undesirable physical health outcomes, there is growing evidence that obesity is also associated with deficits in cognitive

Corresponding author
Andree Hartanto,
andreeh.2014@phdps.smu.edu.sg

functions independent of health conditions that comorbid with obesity (*Whitmer et al., 2008*; *Anstey et al., 2011*; *Beydoun, Beydoun & Wang, 2008*). Two main cognitive aspects tend to be emphasized. First, obesity is associated significantly with impairment of executive functions (*Sellaro & Colzato, 2017*; *Smith et al., 2011*), which are high-level cognitive processes that enable the regulation of thoughts and behavior to achieve goals (*Friedman & Miyake, 2017*). A number of studies have observed that body mass index (BMI) correlates with poorer performance in core components of executive functions, such as inhibitory control, cognitive flexibility, and working memory across different age groups (*Cserjési et al., 2009*; *Goldschmidt et al., 2017*; *Kesse-Guyot et al., 2015*; *Reinert, Po'e & Barkin, 2013*; *Sellaro & Colzato, 2017*; *Smith et al., 2011*). Importantly, the mechanism underlying these relationships is argued to be bidirectional (*Kanoski & Davidson, 2011*; *Sellbom & Gunstad, 2012*). On the one hand, obesity may arise from poor impulse control such that the inability to resist the urge to eat causes overconsumption (*Dohle, Diel & Hofmann, 2018*; *Wu et al., 2016*). On the other hand, emerging animal and human research also indicates that excessive body fat percentages cause neurological changes such as reduced cerebral metabolism, elevated leptin, hippocampal volume reduction, and neuronal degradation (*Erion et al., 2014*; *Raji et al., 2010*; *Smith et al., 2011*; *Volkow et al., 2009*) which can lead to declines in motivation, self-control, and cognition (*Figley et al., 2016*).

A second account links obesity with impairment of episodic memory, which is the ability to store mental representations of events that one has experienced or observed (*Tulving, 2002*). Episodic memory has been shown to play a significant role in the regulation of consumption (*Brunstrom et al., 2012*; *Higgs & Donohoe, 2011*; *Martin, Davidson & McCrory, 2017*). For example, amnesic patients who cannot remember prior episodes of food consumption have been documented to eat several consecutive meals without feeling satiated or uncomfortable (*Hebben et al., 1985*; *Higgs, 2008*). Studies on rodents have found robust evidence for memory deficits which arise from hippocampal abnormalities caused by obesity (*Jurdak, Lichtenstein & Kanarek, 2008*; *Valladolid-Acebes et al., 2011*), thus suggesting that obesity can undermine episodic memory through neurological structural changes the same way that obesity affects brain structure and executive function. However, evidence from studies on humans for any adverse impacts of obesity on episodic memory is far less robust. Although some studies have observed a significant association between higher BMI and poorer episodic memory in young and old adults (*Gunstad et al., 2008*; *Gunstad et al., 2006*), a number of other studies failed to find any such associations (*Conforto & Gershman, 1985*; *Elias et al., 2003*; *Nilsson & Nilsson, 2009*; *Knopman et al., 2001*; *Sabia et al., 2009*; *Singh-Manoux & Marmot, 2005*).

While empirical support for the link between BMI and episodic memory has been difficult to observe, *Cheke, Simons & Clayton (2016)* recently claimed to have found empirical evidence for a direct relationship between higher BMI and poorer episodic memory which was assessed using a Treasure Hunt Task requiring subjects to recall the what, where, and when elements of an episode, thus providing new supporting evidence for the negative relationship between obesity and episodic memory.

Although the study is commendable for utilizing an episodic memory task requiring subjects to engage in contextually rich multidimensional episodic recollection, the study has been criticized most notably by *Cole & Pauly-Takacs (2016)* for a number of limitations, including its relatively small sample size ($n = 60$), failure to control for health conditions that comorbid with obesity (e.g. hypertension, diabetes), and the diminished association between BMI and episodic memory when demographic factors such as age, sex, and education were included in the model. These limitations prompted Cole and Pauly-Takacs to warn that the conclusions are potentially misleading and that the findings should be interpreted with greater caution.

Despite the inconclusive link between obesity and episodic memory, we argue that the inconsistent findings thus far could be due to an overreliance on BMI to operationalize obesity in most studies. Although BMI is currently the most popular index of adiposity, it has received its fair share of criticism (*Gallagher et al., 1996*; *Rothman, 2008*). BMI is calculated as an individual's weight divided by their height squared, but the measurement of weight neither discriminates between muscle and adipose tissue nor does it directly assess regional adiposity (*Stevens, McClain & Truesdale, 2008*), and the measurement of height does not account for the possibility of shrinkage and vertebral collapse (*Price et al., 2006*). *Rothman (2008)* also argued that the use of BMI as a measure of obesity produces a bias towards or away from the null in estimating effects related to obesity. As an alternative, some researchers have found waist-to-hip ratio (WHR) to be superior to BMI, demonstrating in some studies that WHR correlates more highly with health risk factors such as cardiovascular disease, diabetes, and even mortality than BMI does (*Janssen, Katzmarzyk & Ros, 2004*; *Price et al., 2006*; *Zhu et al., 2005*). Thus, re-examinations of obesity and various markers of cognition using alternative indices of adiposity while accounting for the shortcomings of prior research will contribute significantly to the unresolved debate on whether high levels of adiposity are indeed associated with episodic memory deficits in humans.

## The current study

With these issues in mind, the current study sought to re-examine the relationship between obesity and episodic memory while (1) taking into account WHR as a potentially better indicator of obesity and (2) addressing the methodological limitations of previous studies related to sample size, comorbidity, and other possible confounding variables such as demographics and personality. Here, we analysed a large-scale dataset from the Cognitive Project of the National Survey of Midlife Development, Second Wave, in the USA (MIDUS II: Cognitive Project), which offers an opportunity to examine the relationship between obesity and episodic memory with a large sample and WHR as a measure of obesity. Importantly, each participant in the MIDUS II was provided with a tape measure to ensure the accuracy of reported body measurements. Moreover, the study collected detailed information regarding participants' demographics, comorbid health conditions, health behaviors, personality, and even self-perceived obesity, which allowed us to determine whether any associations found between obesity and episodic memory exist above and beyond the influence of these potential confounding variables
(*Bendayan et al., 2017*; *Cole & Pauly-Takacs, 2016*; *Crowe et al., 2010*; *Dunai et al., 2010*; *Hartanto, Toh & Yang, 2016*, *2018*; *Hartanto & Yang, 2018*; *Ihle et al., 2017*; *Williams, Suchy & Kraybill, 2010*). Lastly, participants were administered a comprehensive battery of executive function tasks, which allows us to simultaneously examine the two cognitive functions regarded as most essential to regulating consumption behaviors—executive functions and episodic memory (*Higgs, 2008*).

In summary, using a large sample provided by the MIDUS II, we examined the predictability of BMI and WHR on executive functions and episodic memory after controlling for confounding variables. Specifically, we hypothesized that WHR would be a significant predictor of executive functions and episodic memory, even after controlling for participants' demographics, comorbid health conditions, health behaviors, personality, and even self-perceived obesity. In contrast, based on the shortcomings of BMI as a measure of obesity, we hypothesized that BMI would not be a significant predictor of executive functions and episodic memory especially when confounding factors are taken into account.

## METHOD

### Participants

The sample consisted of 4,206 participants from the Cognitive Project of the National Survey of Midlife Development, Second Wave, in the USA (MIDUS II: Cognitive Project; *Ryff & Lachman, 2010*). MIDUS II was conducted in 2004–2006 on a nationally representative random-digit-dial sample of non-institutionalized, English-speaking adults (see *Ryff et al., 2007* for more details). As the self-administered questionnaire contained the key predictors (BMI and WHR) and most of the covariates (e.g. health status, personality traits, etc.) of interest to the current study, we excluded participants who did not complete the self-administered questionnaire ($n = 494$). After exclusions, the final sample size was 3,712. Table 1 summarized the demographic, health-related, and personality characteristics of the sample. The data collection for the MIDUS project was approved by the Education and Social/Behavioral Sciences and the Health Sciences Institutional Review Board at the University of Wisconsin-Madison (H-2008-0060). All participants have provided informed consent. Data and materials from the MIDUS II: Cognitive Project are freely available from the Inter-University Consortium for Political and Social Research (http://www.icpsr.umich.edu).

### Measures

#### Cognitive ability

Episodic memory and executive functions were assessed with the brief test of adult cognition by telephone (BTACT; *Lachman & Tun, 2008*; *Tun & Lachman, 2006*). The BTACT is a battery of cognitive function tests which comprise the Immediate Word List Recall Task, Backward Digits Span, Categorical Fluency, Stop and Go Switch Task (SGST), Number Series, Backward Counting Task, and Delayed Word List Recall (see Table 2 for a summary of these cognitive function tests). Exploratory and confirmatory factor analyses of the seven cognitive tests show that the data fits a two-factor model of episodic memory and executive functions (*Lachman & Tun, 2008*).

**Table 1 Descriptive statistics for obesity, cognitive functions, demographics, health status, health behaviors, and personality characteristics.**

|  | M | SD | Range |
|---|---|---|---|
| **Demographic** |  |  |  |
| Mean age (years) | 56.49 | 12.35 | 32–84 |
| Sex (% of male) | 45% |  |  |
| Education[1] | 7.28 | 2.54 | 1–12 |
| Household income ($) | 57,081 | 55,217 | 0–200,000 |
| Subjective social status | 4.49 | 1.82 | 1–10 |
| **Health status** |  |  |  |
| Number of chronic disease[2] | 2.44 | 2.38 | 0–13 |
| Stroke | 1% |  |  |
| Hypertension | 31% |  |  |
| Diabetes | 10% |  |  |
| Physical health evaluation | 2.44 | 1.01 | 1–5 |
| **Health behaviors** |  |  |  |
| Non-smoker (%) | 52% |  |  |
| Former smoker (%) | 34% |  |  |
| Current smoker (%) | 14% |  |  |
| Alcohol consumer (%) | 59% |  |  |
| **Personality[3]** |  |  |  |
| Neuroticism | 2.06 | 0.62 | 1–4 |
| Openness to experience | 2.90 | 0.54 | 1–4 |
| Conscientious | 3.46 | 0.45 | 1–4 |
| Extraversion | 3.10 | 0.57 | 1–4 |
| Agreeableness | 3.45 | 0.50 | 1–4 |
| **Obesity** |  |  |  |
| Body mass index[2] | 27.89 | 5.67 | 14.23–51.10 |
| Waist-to-hip ratio[2] | 0.91 | 0.11 | 0.50–1.32 |
| **Cognitive functions** |  |  |  |
| Episodic memory (z-score) | 0.01 | 1.01 | −3.07 to 3.83 |
| Executive functions (z-score) | 0.00 | 0.99 | −4.80 to 3.39 |

**Notes:**
SDs are shown in parentheses.
[1] Education attainment was rated on a scale of 1 (*No school*) to 12 (*Ph.D, Ed.D, MD, LLB, LLD, JD, or other professional degree*).
[2] Values were winsorized to reduce the effect of extreme outliers.
[3] Each personality score was computed by averaging the respective personality items (*Rossi, 2001*) rated on a four-point Likert scale (1 = *not at all,* 4 = *a lot*), with higher scores indicating a higher amount of that particular personality dimension (e.g. greater neuroticism).

Episodic memory is best represented by performance on the immediate word list recall and delayed word list recall while executive function is best represented by performance on the backward digit span, categorical fluency, number series, backward counting, and SGST. Following the recommendations of Lachman and Tun, we computed composite scores for episodic memory and executive functions from standardized scores of the respective subtests, with a mean of zero and a standard deviation of one (see *Lachman & Tun, 2008* for more details).

**Table 2 Overview and descriptive statistics of cognitive function measures.**

| Measures | Theoretical construct | Performance index | Procedure | *M* (SD) | Range |
|---|---|---|---|---|---|
| Immediate word list recall task | Episodic verbal memory | Total number of accurate responses | Participants were presented with 15 words and instructed to recall the items | 6.74 (2.30) | 0–15 |
| Backward digit span | Working memory | Highest number of digits recalled up to 8 | Participants were given a series of numbers and instructed to recall the numbers backwards | 5.01 (1.50) | 0–8 |
| Categorical fluency | Verbal ability and speed of processing | The frequency of unique responses generated | Participants were instructed to produce as many words as possible from each given category within a minute | 18.80 (6.17) | 0–42 |
| Delayed word list recall task | Episodic verbal memory and forgetting | Total number of accurate responses | Participants were presented with 15 words and instructed to recall the items after a long delay | 4.45 (2.63) | 0–14 |
| Number series | Inductive reasoning and fluid intelligence | Total number of accurate responses | Participants were given strings of numbers and instructed to deduce the next number in the series | 2.28 (1.52) | 0–5 |
| Backward counting task | Speed of processing | Total number of accurate responses, derived from the last number reached, taking off from errors | Participants were instructed to count backwards from 100 | 37.15 (11.31) | −2 to 90 |
| Stop and go switch task | Inhibitory control and task-switching | The average of reaction time differences between incongruent trials and congruent trials (inhibitory control) and reaction time differences between switch trials and nonswitch trials (task-switching) | Participants were instructed to complete two single-task blocks and a mixed-task block. In the first single-task block, all of the trials were congruent trials where participants were instructed to give a verbal response 'stop' and 'go' as quickly as possible when they heard the words 'red' and 'green,' respectively. In the second single-task block, all of the trials were incongruent where participants were instructed to respond with 'go' and 'stop' when they heard 'red' and green,' respectively. In the mixed block, participants were required to alternate between congruent and incongruent rules based on the cue presented ('normal' or 'reverse') by the experimenter | 1.09 (0.25) | 0.22–3.82 |

## Obesity

Obesity was indexed by BMI and WHR. BMI was calculated by participants' self-reported weight and height based on the formula where BMI equals to kilograms per meters squared. Participants' WHR was calculated by the ratio of their waist around the navel to their hips at the widest point. During the data collection, participants were provided with a tape measure to ensure the accuracy of the reported body measurements. Participants were instructed to stand upright and keep the tape measure taut to the body when making the measurements (e.g. avoid draping the tape measure loosely over

their clothing). Participants were also specifically instructed to measure at the level of their navel for the waist measurement and at the widest point between their waist and thighs for the hip measurement. Following the recommendations of *Preston, Fishman & Stokes (2015)*, recommendations, we operationalized BMI and WHR as a continuous variable to minimize bias associated with treating obesity as a categorical variable while keeping the operationalization of both BMI and WHR consistent.

## Data analysis

We examined the influence of BMI and WHR on episodic memory and executive functions. For each criterion, we performed ordinary least squares regression models with BMI and WHR as separate predictors to minimize multicollinearity.
Each predictor–criterion pair consisted of four models, with each model consisting of an additional set of covariates that have been linked to episodic memory and executive functions to ensure the robustness of the hypothesized associations. Our analyses were performed using IBM SPSS Statistics Version 25.

In the first model, we included participants' demographics such as age, sex, education attainment, household income, and subjective social status as measured by the MacArthur Scales of Subjective Social Status (*Adler et al., 2000*). Education attainment was rated on a scale of 1 (*No school*) to 12 (*Ph.D, Ed.D, MD, LLB, LLD, JD, or other professional degree*). Socioeconomic status indices, such as education attainment, household income, and subjective social status, have been shown to be significant and thus important predictors of interindividual variability in cognitive functions (*Gianaros et al., 2007*; *Hackman, Farah & Meaney, 2010*; *Zhang, Fung & Kwok, 2017*). In the second model, we included comorbidities of obesity that have been found to be associated with cognitive decline, such as hypertension, diabetes, and stroke (*Crowe et al., 2010*; *Ihle et al., 2017*). We also controlled for self-reported physical health and the total number of chronic diseases experienced in the past 12 months as general indicators of health status (*Bendayan et al., 2017*). In addition, we included health-impacting behaviors such as smoking, alcohol consumption, and physical activity. Following *Lee (2014)*, physical activity was based on the average of four self-reported items that assessed the frequency of vigorous and moderate physical activity in both summer and winter seasons for example 'How often do you engage in vigorous physical activity that causes your heart to beat so rapidly that you can feel it in your chest and you perform the activity long enough to work up a good sweat and are breathing heavily? (Examples: competitive sports like running, vigorous swimming, or high intensity aerobics; digging in the garden, or lifting heavy objects).' In the third model, we included the Big Five personality traits as covariates (extraversion, conscientiousness, agreeableness, neuroticism, and openness to experience; *Rossi, 2001*) to ensure that individual personality characteristics are not a confounding factor (*Williams, Suchy & Kraybill, 2010*). In the last model, we included self-perceived obesity as a covariate to rule out the possibility that the hypothesized associations might be due to psychological issues associated with how obese participants believed themselves to be (e.g. body dysmorphic disorder, low self-confidence) rather than actual physical obesity

(*Dunai et al., 2010*). We further conducted slope differentiation tests to examine the predictability differences between BMI and WHR on episodic memory and executive functions.

In addition, we conducted moderation analyses separately to examine whether age moderates the predictability of BMI and WHR. Age was explored as a moderator due to the possibility that the lack of a relationship between BMI and cognitive functions is limited only to older adults, as BMI may not be sensitive to shrinkage and vertebral collapse as a function of age (*Price et al., 2006*). Here, we reconducted our analyses by including the interaction terms between BMI and age or WHR and age in each model. Furthermore, given that males and females differ in adipose tissue distribution and expansion (*Zore, Palafox & Reue, 2018*), we also conducted separate analyses to examine whether the predictability of BMI and WHR would be moderated by sex. At the end of our analyses, we also conducted polynomial regressions for each model to examine the quadratic influence of BMI and WHR on episodic memory and executive functions due to the possible cognitive consequences of extremely low weight individuals.

In each model, BMI, WHR, and number of chronic conditions were winsorized to minimize the influence of outliers. Age, education, household income, subjective social status, number of chronic diseases, the Big Five personality traits, and self-perceived obesity were mean-centred to improve the interpretation of intercept terms. In our moderation analyses, BMI and WHR were mean-centred. For missing data in our datasets, we performed multiple imputations (*Rubin, 1987*) using a Markov chain Monte Carlo algorithm with a fully conditional specification procedure to create five imputed datasets. As recommended by *Von Hippel (2007)*, we employed the multiple-imputation-then-deletion procedure in which missing criterion variables were excluded from the analysis subsequent to the imputation. Collinearity statistics did not indicate multicollinearity and none of the predictors and covariates of our regression models had tolerance values lower than 0.10 or variance inflation factor values larger than 10 (*O'brien, 2007*; *York, 2012*).

# RESULTS

## Body mass index

Our models for the effect of BMI on episodic memory and executive functions are summarized in Table 3. After controlling for demographic variables in the first model, we observed that BMI was negatively associated with episodic memory ($B = -0.007$, SE = 0.003, 95% CI [$-0.012$, $-0.002$], $p = 0.009$). However, this significant association diminished when we included health status and health behaviors as covariates in the second model ($B = -0.001$, SE = 0.003, 95% CI [$-0.007$, 0.004], $p = 0.640$), personality variables in the third model ($B = -0.002$, SE = 0.003, 95% CI [$-0.007$, 0.004], $p = 0.567$), and self-perceived obesity in the fourth model ($B = -0.002$, SE = 0.004, 95% CI [$-0.010$, 0.006], $p = 0.617$).

Similar with the results for episodic memory, we also observed that BMI was negatively associated with executive functions after controlling for demographic variables in the first model ($B = -0.007$, SE = 0.002, 95% CI [$-0.011$, $-0.002$], $p = 0.007$), but the significant

**Table 3** Model summaries of episodic memory and executive functions with BMI as the predictor.

| | Episodic memory (n = 3,700) | | | | Executive functions (n = 3,707) | | | |
|---|---|---|---|---|---|---|---|---|
| | Model 1 | Model 2 | Model 3 | Model 4 | Model 1 | Model 2 | Model 3 | Model 4 |
| **Predictor** | | | | | | | | |
| BMI | −0.039* | −0.008 | −0.009 | −0.011 | −0.037* | 0.012 | 0.009 | −0.025 |
| **Covariates** | | | | | | | | |
| Age | −0.295** | −0.257** | −0.268** | −0.268** | −0.341** | −0.295** | −0.302** | −0.302** |
| Sex | −0.239** | −0.246** | −0.242** | −0.241** | 0.070** | 0.060** | 0.058** | 0.066** |
| Education | 0.173** | 0.150** | 0.145** | 0.145** | 0.332** | 0.291** | 0.280** | 0.279** |
| Household income | 0.035† | 0.017 | 0.020 | 0.020 | 0.090** | 0.057* | 0.060* | 0.058* |
| Subjective status | −0.012 | 0.000 | 0.024 | 0.024 | −0.004 | 0.020 | 0.022 | 0.023 |
| Hypertension | | −0.007 | −0.009 | −0.009 | | −0.005 | −0.005 | −0.005 |
| Diabetes | | −0.011 | −0.014 | −0.014 | | −0.015 | −0.015 | −0.014 |
| Stroke | | −0.020 | −0.019 | −0.019 | | −0.039* | −0.040* | −0.039* |
| Self-rated health | | −0.062** | −0.048* | −0.048* | | −0.119** | −0.118** | −0.119** |
| Chronic disease | | −0.008 | 0.005 | 0.005 | | −0.024 | −0.017 | −0.019 |
| Former smoker | | 0.013 | 0.014 | 0.014 | | 0.027† | 0.027† | 0.025† |
| Current smoker | | 0.031† | 0.029 | 0.029 | | 0.004 | −0.003 | −0.004 |
| Alcohol | | 0.023 | 0.023 | 0.023 | | 0.041* | 0.043* | 0.041* |
| Physical activity | | 0.084** | 0.082** | 0.082** | | 0.098** | 0.101** | 0.100 |
| Agreeableness | | | 0.014 | 0.014 | | | 0.027 | 0.025 |
| Openness | | | 0.028 | 0.028 | | | 0.025 | 0.028† |
| Neuroticism | | | −0.054* | −0.054* | | | −0.044* | −0.044* |
| Extraversion | | | 0.000 | 0.000 | | | −0.067** | −0.067* |
| Conscientiousness | | | 0.025 | 0.025 | | | 0.007 | 0.007 |
| Perceived obesity | | | | 0.002 | | | | 0.049* |
| **Model statistics** | | | | | | | | |
| R | 0.447 | 0.462 | 0.467 | 0.467 | 0.577 | 0.604 | 0.608 | 0.609 |
| R² | 0.200 | 0.213 | 0.219 | 0.219 | 0.333 | 0.365 | 0.370 | 0.371 |
| R² change | 0.200** | 0.014** | 0.005** | 0.000 | 0.333** | 0.033** | 0.004** | 0.001* |

Notes:
BMI, age, sex, education, household income, and subjective socioeconomic status (SES) were included in the Model 1. Hypertension, diabetes, stroke, self-reported physical health, the total number of chronic diseases, smoking, alcohol consumption, and physical activity were further included as covariates in the Model 2. Big five personality traits, including extraversion, conscientiousness, agreeableness, neuroticism, and openness to experience, were further included as covariates in the Model 3. Self-perceived obesity was additionally included as a covariate in the Model 4. Sex was coded with female as reference. Former and current smoker were coded with non-smoker as reference. Regular alcohol intake was coded with non-drinker as reference. Hypertension, diabetes, and stroke were coded with no respective health condition experienced for the last 12 months as reference. Model statistics were based on the first imputed dataset.
† $p < 0.10$.
* $p < 0.05$.
** $p < 0.001$.

association diminished when we included health status and health behaviors as covariates in the second model (B = 0.002, SE = 0.003, 95% CI [−0.003, 0.007], $p = 0.422$), personality variables in the third model (B = 0.002, SE = 0.003, 95% CI [−0.003, 0.007], $p = 0.545$), and self-perceived obesity in the fourth model (B = −0.004, SE = 0.004, 95% CI [−0.011, 0.002], $p = 0.204$). These results suggest that the significant prediction of BMI on episodic memory and executive functions in the first model could be driven by health issues comorbid with obesity that have been associated with cognitive decline.

**Table 4 Model summaries of episodic memory and executive functions with waist-to-hip ratio as the predictor.**

| | Episodic memory (n = 3,700) | | | | Executive functions (n = 3,707) | | | |
|---|---|---|---|---|---|---|---|---|
| | Model 1 | Model 2 | Model 3 | Model 4 | Model 1 | Model 2 | Model 3 | Model 4 |
| **Predictor** | | | | | | | | |
| Waist-to-hip ratio | −0.069** | −0.049* | −0.047* | −0.047* | −0.091** | −0.059* | −0.060** | −0.070** |
| **Covariates** | | | | | | | | |
| Age | −0.293** | −0.257** | −0.268** | −0.267** | −0.338** | −0.300** | −0.306** | −0.301** |
| Sex | −0.202** | −0.219** | −0.215** | −0.215** | 0.119** | 0.094** | 0.094** | 0.104** |
| Education | 0.172** | 0.148** | 0.144** | 0.144** | 0.329** | 0.288** | 0.277** | 0.277** |
| Household income | 0.032$^{†}$ | 0.015 | 0.018 | 0.018 | 0.085** | 0.055* | 0.058** | 0.055* |
| Subjective status | −0.011 | 0.001 | 0.025 | 0.025 | −0.002 | 0.021 | 0.022 | 0.023 |
| Hypertension | | −0.006 | −0.009 | −0.009 | | 0.000 | −0.001 | −0.006 |
| Diabetes | | −0.010 | −0.014 | −0.014 | | −0.011 | −0.012 | −0.015 |
| Stroke | | −0.020 | −0.019 | −0.018 | | −0.040* | −0.040* | −0.038* |
| Self-rated health | | −0.060* | −0.046* | −0.047* | | −0.113** | −0.113** | −0.118** |
| Chronic disease | | −0.007 | 0.006 | 0.006 | | −0.020 | −0.015 | −0.018 |
| Former smoker | | 0.013 | 0.014 | 0.014 | | 0.026$^{†}$ | 0.027$^{†}$ | 0.025$^{†}$ |
| Current smoker | | 0.032* | 0.030$^{†}$ | 0.031$^{†}$ | | 0.003 | 0.003 | 0.007 |
| Alcohol | | 0.022 | 0.022 | 0.022 | | 0.039* | 0.041* | 0.041* |
| Physical exercise | | 0.082** | 0.080** | 0.080** | | 0.094** | 0.098** | 0.098** |
| Agreeableness | | | 0.016 | 0.015 | | | 0.030$^{†}$ | 0.027 |
| Openness | | | 0.027 | 0.027 | | | 0.024 | 0.026 |
| Neuroticism | | | −0.053* | −0.053* | | | −0.044* | −0.043* |
| Extraversion | | | −0.001 | −0.001 | | | −0.069** | −0.068** |
| Conscientiousness | | | 0.024 | 0.023 | | | 0.004 | 0.005 |
| Perceived obesity | | | | 0.003 | | | | 0.044* |
| **Model statistics** | | | | | | | | |
| R | 0.449 | 0.464 | 0.469 | 0.469 | 0.580 | 0.606 | 0.610 | 0.611 |
| $R^2$ | 0.202 | 0.215 | 0.220 | 0.220 | 0.337 | 0.367 | 0.372 | 0.373 |
| $R^2$ change | 0.202** | 0.014** | 0.005** | 0.000** | 0.337** | 0.031** | 0.005** | 0.002* |

Notes:

Waist-to-hip ratio, age, sex, education, household income, and subjective socioeconomic status (SES) were included in the Model 1. Hypertension, diabetes, stroke, self-reported physical health, the total number of chronic diseases, smoking, alcohol consumption, and physical activity were further included as covariates in the Model 2. Big five personality traits, including extraversion, conscientiousness, agreeableness, neuroticism, and openness to experience, were further included as covariates in the Model 3. Self-perceived obesity was additionally included as a covariate in the Model 4. Sex was coded with female as reference. Former and current smoker were coded with non-smoker as reference. Regular alcohol intake was coded with non-drinker as reference. Hypertension, diabetes, and stroke were coded with no respective health condition experienced for the last 12 months as reference. Model statistics were based on the first imputed dataset.

$^{†}$ $p < 0.10$.
$^{*}$ $p < 0.05$.
$^{**}$ $p < 0.001$.

## Waist-to-hip ratio

Our models for the effect of WHR on episodic memory and executive functions are summarized in Table 4. Similar with BMI, we observed that WHR was negatively associated with episodic memory after controlling for demographic variables in the first model ($B = −0.657$, SE = 0.179, 95% CI [−1.009, −0.305], $p < 0.001$). However, in contrast to BMI, WHR remained a significant predictor of episodic memory after we included health status and health behaviors as covariates in the second model
($B = -0.469$, SE = 0.179, 95% CI [−0.821, −0.117], $p = 0.009$), personality variables in the third model ($B = -0.447$, SE = 0.180, 95% CI [−0.800, −0.095], $p = 0.013$), and self-perceived obesity in the fourth model ($B = -0.454$, SE = 0.183, 95% CI [−0.814, −0.095], $p = 0.013$). When similar analyses were performed for executive functions, we also found that WHR was a significant predictor of executive functions after we controlled for demographic variables in the first model ($B = -0.856$, SE = 0.161, 95% CI [−1.172, −0.540], $p < 0.001$), health status and health behaviors in the second model ($B = -0.659$, SE = 0.160, 95% CI [−0.869, −0.241], $p = 0.001$), personality variables in the third model ($B = -0.565$, SE = 0.160, 95% CI [−0.880, −0.250], $p < 0.001$), and self-perceived obesity in the fourth model ($B = -0.659$, SE = 0.163, 95% CI [−0.980, −0.339], $p < 0.001$).

We further conducted slope differentiation tests between BMI and WHR. Crucially, in all models we examined, we consistently found significantly lower coefficient estimates for BMI than WHR ($ps < 0.05$), suggesting that WHR is related more strongly to episodic memory and executive functions than BMI. Taken together, our results demonstrate that higher WHR is associated with poorer episodic memory and executive functions, and that this effect persists beyond the influence of demographics, comorbid health issues, health status, health behaviors, personality traits, and self-perceived obesity.

### Age as moderator

In our additional moderation analyses for BMI, we did not observe any significant interactions between BMI and age in predicting episodic memory in all of our models; Model 1 (95% CI [−0.001, 0.000]; $\beta = -0.009$, $p = 0.553$), Model 2 (95% CI [−0.001, 0.000], $\beta = -0.009$, $p = 0.437$), Model 3 (95% CI [−0.001, 0.000], $\beta = -0.012$, $p = 0.422$), Model 4 (95% CI [−0.001, 0.000], $\beta = -0.012$, $p = 0.415$). There was also no significant interaction between BMI and age in predicting executive functions in Model 1 (95% CI [0.000, 0.001]; $\beta = 0.018$, $p = 0.208$), Model 2 (95% CI [0.000, 0.001], $\beta = 0.015$, $p = 0.274$), Model 3 (95% CI [0.000, 0.001], $\beta = 0.015$, $p = 0.265$), and Model 4 (95% CI [0.000, 0.001], $\beta = 0.012$, $p = 0.375$).

Similarly, in our additional moderation analyses for WHR, the interaction term between WHR and age did not significantly predict episodic memory in any of the four models; Model 1 (95% CI [−0.044, 0.002]; $\beta = -0.027$, $p = 0.071$), Model 2 (95% CI [−0.045, 0.000], $\beta = -0.029$, $p = 0.051$), Model 3 (95% CI [−0.044, 0.001], $\beta = -0.028$, $p = 0.065$), Model 4 (95% CI [−0.044, 0.001], $\beta = -0.028$, $p = 0.065$). Similarly, the interaction term between WHR and age did not significantly predict executive functions in any of the models; Model 1 (95% CI [−0.032, 0.010]; $\beta = -0.014$, $p = 0.307$), Model 2 (95% CI [−0.033, 0.008], $\beta = -0.016$, $p = 0.242$), Model 3 (95% CI [−0.032, 0.008], $\beta = -0.016$, $p = 0.251$), Model 4 (95% CI [−0.032, 0.008], $\beta = -0.016$, $p = 0.244$). These findings indicate that the superiority of WHR over BMI as a predictor of episodic memory and executive functions is consistent and robust across adulthood.

### Sex as moderator

Likewise, no significant interactions between BMI and sex were observed when predicting episodic memory across our four models; Model 1 (95% CI [−0.010, 0.012]; $\beta = 0.005$,

$p = 0.797$), Model 2 (95% CI [−0.009, 0.013], $\beta = 0.006$, $p = 0.749$), Model 3 (95% CI [−0.010, 0.012], $\beta = 0.004$, $p = 0.819$), Model 4 (95% CI [−0.010, 0.012], $\beta = 0.004$, $p = 0.825$). There was also no significant interaction between BMI and sex in predicting executive functions in Model 1 (95% CI [−0.018, 0.002]; $\beta = -0.026$, $p = 0.123$), Model 2 (95% CI [−0.017, 0.002], $\beta = -0.025$, $p = 0.128$), Model 3 (95% CI [−0.017, 0.002], $\beta = -0.024$, $p = 0.137$), and Model 4 (95% CI [−0.018, 0.001], $\beta = -0.027$, $p = 0.094$).

Neither did sex interact with WHR when predicting episodic memory across any of the four models; Model 1 (95% CI [−0.372, 0.984]; $\beta = 0.021$, $p = 0.376$), Model 2 (95% CI [−0.406, 0.958], $\beta = 0.019$, $p = 0.427$), Model 3 (95% CI [−0.488, 0.876], $\beta = 0.013$, $p = 0.577$), Model 4 (95% CI [−0.487, 0.877], $\beta = 0.013$, $p = 0.575$). The interaction term between WHR and sex also did not significantly predict executive functions in any of the models; Model 1 (95% CI [−0.027, 1.201]; $\beta = 0.040$, $p = 0.061$), Model 2 (95% CI [−0.106, 1.092], $\beta = 0.034$, $p = 0.107$), Model 3 (95% CI [−0.189, 1.008], $\beta = 0.028$, $p = 0.180$), Model 4 (95% CI [−0.177, 1.015], $\beta = 0.029$, $p = 0.169$). The results suggest that the stronger predictability of WHR relative to BMI on episodic memory and executive functions is consistent for both men and women.

### Polynomial regressions

Lastly, we conducted polynomial regressions to examine the quadratic relations of BMI and WHR in predicting episodic memory and executive functions in the four different models. For BMI, the quadratic relations between BMI and episodic memory were not significant across all of the four models after controlling for demographic variables in Model 1 (95% CI [−0.001, 0.000], $p = 0.441$), health status and health behaviors in Model 2 (95% CI [−0.001, 0.000], $p = 0.646$), personality variables in Model 3 (95% CI [−0.001, 0.000], $p = 0.604$), and self-perceived obesity in Model 4 (95% CI [−0.001, 0.000], $p = 0.611$). There were also no significant quadratic relations between BMI and executive functions in any of the models; Model 1 (95% CI [−0.001, 0.000], $p = 0.488$), Model 2 (95% CI [−0.001, 0.000], $p = 0.976$), Model 3 (95% CI [−0.001, 0.000], $p = 0.891$), Model 4 (95% CI [0.000, 0.001], $p = 0.424$).

Similarly, we did not observe any significant quadratic relations between WHR and episodic memory in any of the models; Model 1 (95% CI [−0.267, 2.665], $p = 0.109$), Model 2 (95% CI [−0.461, 2.466], $p = 0.179$), Model 3 (95% CI [−0.566, 2.358], $p = 0.229$), and Model 4 (95% CI [−0.555, 2.375], $p = 0.223$). We also did not observe any significant quadratic relations between WHR and executive functions in Model 1 (95% CI [−0.673, 1.943], $p = 0.341$); Model 2 (95% CI [−1.004, 1.573], $p = 0.665$), Model 3 95% CI [−1.091, 1.485], $p = 0.764$), and Model 4 (95% CI [−0.957, 1.627], $p = 0.611$). These findings suggest that the relations between obesity and cognitive functions are unlikely to be quadratic.

## DISCUSSION

Research on adiposity and cognition has routinely found associations between obesity and executive functions, but empirical support for a relationship between obesity and episodic memory has been less forthcoming. While *Cheke, Simons & Clayton (2016)*

presented a decent effort to investigate the obesity-memory link, their study received criticism for having various issues including small sample size, not controlling for health conditions that comorbid with obesity, and the diminished association between BMI and episodic memory when demographic variables were included in the model (*Cole & Pauly-Takacs, 2016*). In consideration of these issues and also noting the problems associated with using BMI as a measure of adiposity (cf., *Rothman, 2008*), we sought to re-examine the relationship between obesity and episodic memory using data from the Cognitive Project of the MIDUS II. The MIDUS II is a large-scale study which comprised numerous participants, assessed participants' BMI, WHR, executive functions, and episodic memory, and included numerous measures of potential confounding variables, thus allowing us to simultaneously address the aforementioned shortcomings of prior research.

Our analyses showed that WHR is a superior alternative to BMI as a measure of the adverse impact of obesity on episodic memory as well as executive functions. Although BMI rivalled WHR in predicting executive functions and episodic memory while controlling for demographic variables, the effect of BMI on executive functions and episodic memory diminished after health status, health behaviors, personality variables, and self-perceived obesity were added to the model. In contrast, these additional variables did not confound the effect of WHR on episodic memory and executive functions. The current study therefore contributes significantly to the debate on obesity and cognition by demonstrating that WHR surpasses BMI in predicting episodic memory and executive functions across adulthood, at least based on the MIDUS II dataset.

Importantly, the current study also demonstrated a clear link between deficits in episodic memory and higher levels of adiposity as indexed by WHR. By carefully controlling for a wide range of possible confounding factors such as demographic, health-related, personality, and self-perception variables, we found that WHR maintained a distinct effect on episodic memory, thus indicating that obesity is linked to people's ability to store and recall mental representations of experienced or observed events independent of lifestyle or other factors. Our results suggest that not only can impaired episodic memory cause people to overconsume because they fail to recollect prior episodes of eating, but being overweight can also possibly impair episodic memory, which in turn further exacerbates overconsumption. These findings lend support to 'vicious cycle' models of obesity and cognitive decline (*Kanoski & Davidson, 2011*; *Sellbom & Gunstad, 2012*), where high levels of adiposity contribute to cognitive processes that induce ever higher levels of adiposity. In light of our findings, health practitioners should take into account the possibly greater difficulty experienced by overweight individuals relative to non-overweight individuals in regulating consumption behaviors and tailor treatment programmes according to the specific needs of overweight individuals.

Our results increase the confidence that continued probing into this area will eventually unearth the specific biological mechanisms that underlie the associations between obesity and episodic memory as well as other aspects of executive function. One candidate mechanism of impaired cognitive function resulting from the accumulation of visceral fat is the secretion of pro-inflammatory cytokines which can result in insulin resistance

and worsen cerebrovascular reactivity (*Adabimohazab et al., 2016*; *Lambert et al., 2015*; *Miller & Spencer, 2014*). Another possible mechanism for the impairment of cognitive functioning is central and adipose inflammation which plays a significant role in reduced synaptic plasticity (*Erion et al., 2014*; *Willette & Kapogiannis, 2015*). Various researchers are also especially interested in the psychological and behavioral processes that drive impaired cognition and overconsumption, such as whether successful recollection of prior episodes of behavior is hindered by deficits in encoding or retrieval processes (*Cheke, Simons & Clayton, 2016*; *Sohrabi et al., 2015*). Future studies that assess the specific elements of episodic memory deficits are thus needed so that therapists and health practitioners may streamline their treatments more specifically to target patient behavioral deficiencies.

It is important to note some limitations of the current study. Although we were able to rule out a large number of confounding factors and achieve higher confidence in the direct impact of obesity (indexed by BMI and WHR) on episodic memory and executive functions, the findings are correlational and cross-sectional and therefore limit our ability to make causal claims. Further research, in particular longitudinal studies, are warranted to assess the nuanced effects of obesity on cognition and vice versa as these effects unfold over time.

There may be some concern over the validity of the self-reported BMI in our study. However, we believe that this is unlikely to compromise the overall results because self-reported height and weight have been documented to be adequately reliable and valid in empirical studies on adult samples, even after demographic variables have been controlled (*Spencer et al., 2002*; *Stommel & Schoenborn, 2009*). Furthermore, a subset of participants from the MIDUS II study also participated in a biomarker project where professional clinicians collected their height and weight information (*Love et al., 2010*). The self-reported BMI of the MIDUS II Cognitive Project correlated very strongly with the professionally measured BMI of the MIDUS II Biomarker Project ($r = 0.922$, $p < 0.001$), thus indicating that the self-reported BMI in the current dataset is probably valid.

Due to the lack of available information on adiposity fat percentage and dietary habits in the MIDUS II dataset, we were unable to assess the effects of adiposity fat percentage (*Kamijo et al., 2012*) nor were we able to include diet as potential covariates (*Kanoski & Davidson, 2011*; *Yeomans, 2017*), both of which have been shown to be important factors associated with cognitive functioning. Nonetheless, future studies that measure the percentage of adiposity fat when examining the relationship between obesity and cognitive functions while accounting for dietary habits will likely prove promising.

## CONCLUSIONS

Taken together, the findings of our study reveal that obesity is not only associated with deficits in executive functions but also episodic memory. However, the adverse links between obesity and cognitive functions only emerged when obesity was measured by WHR and not BMI, highlighting the need for future studies to use more sensitive measures of obesity when examining its relationship with individual differences in cognitive functions.

### Funding

The MIDUS I study was supported by a grant from the John D. Catherine T. MacArthur Foundation Research Network on Successful Midlife Development. The MIDUS II study was supported by a grant from the National Institute on Aging (P01-AG020166) to conduct a longitudinal follow-up of the MIDUS I study. The funders had no role in study design, data collection and analysis, decision to publish, or preparation of the manuscript.

### Grant Disclosures

The following grant information was disclosed by the authors:
Foundation Research Network on Successful Midlife Development.
National Institute on Aging: P01-AG020166.

### Competing Interests

The authors declare that they have no competing interests.

### Author Contributions

- Andree Hartanto conceived and designed the experiments, analysed the data, contributed reagents/materials/analysis tools, prepared figures and/or tables, authored or reviewed drafts of the paper, approved the final draft.
- Jose C. Yong conceived and designed the experiments, authored or reviewed drafts of the paper, approved the final draft.

### Human Ethics

The following information was supplied relating to ethical approvals (i.e., approving body and any reference numbers):

The data collection for the MIDUS project was approved by the Education and Social/Behavioral Sciences and the Health Sciences Institutional Review Board at the University of Wisconsin-Madison (H-2008-0060).

### Data Availability

Data and materials from the MIDUS II: Cognitive Project are freely available from the Inter-University Consortium for Political and Social Research (https://www.icpsr.umich.edu/icpsrweb/ICPSR/studies/29282).

### Supplemental Information

Supplemental information for this article can be found online at http://dx.doi.org/10.7717/peerj.5624#supplemental-information.

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
