# Peer review of "Measurement matters: higher waist-to-hip ratio but not body mass index is associated with deficits in executive functions and episodic memory"

_PeerJ, doi:10.7717/peerj.5624_

## Round 0.1 · original submission · Major Revisions

I now have received reviewers' comments. Although reviewers pointed out some strength in your study, several aspects of this manuscript should be revised to improve its clarity. Their observations are presented with clarity so I'll not risk confusing matters by belaboring or reiterating their comments. While I might quibble with the occasional point, I note that I regard the reviewers' opinions as substantive and well-informed. I believe that all of the highlighted reservations require contemplation and appropriate attention in revising the document if it is to contribute appropriately to Peerj and the extant literature. Please revise or refute according to the two reviewers' comments and provide a point by point reply in addition to the revised manuscript.

Tsung-Min Hung, PhD., FNAK, PeerJ editor
Research chair professor, Department of Physical Education, National Taiwan Normal University

Reviewer 1 ·

Basic reporting

No comment.

Experimental design

The biggest shortcoming of the study was that BMI was calculated by participants’ self-reported weight and height. This may be one of the reasons that there was no relationship between BMI and cognitive outcomes (EF and episodic memory).

In addition, please clarify more details of the methods of measuring WHR.
Who measured the WHR of the participants?

Validity of the findings

Another major concern is the effect of multicollinearity.
Please provide the variance inflation factor (VIF).

I recommend adding scatter plot of main results.

In addition, why did authors not conduct moderation analyses that examine whether the relationship of obesity to EF and episodic memory could be moderated by sex?
I recommend the moderation analyses for the following reasons.
1. Gender differences in the regulation of body-weight are well documented.
2. Sex may be a biological variable that related to cognitive decline (e.g., Sohn et al., 2018, Scientific Reports).
3. Males and females differ in adipose tissue distribution and expansion (Zore et al., 2018, Molecular Metabolism, for a review).

·

Basic reporting

Introduction-
1. The Intro is well-written and easy-to-follow except that a hypothesis regarding the association of BMI/WHR with executive function/episodic memory should be provided.
2. P 4, line 85: this reference is old; maybe a newer one?

Results-
1. Some relevant information regarding regression analysis is missing. Please provide data on model summary, including model r2, model r2 change, and model P ANOVA.
2. Please provide table summarizing task performance on executive function and episodic memory.

Discussion-
Currently, the Intro and Discussion are unbalanced. The reviewer suggests the authors expand the discussion. The reviewer suggests the authors elaborate the possible mechanisms underlie the association between obesity and executive function/episodic memory. For example, it has been documented that accumulation of visceral fat may induce secretion of pro-inflammatory cytokines, which thereby results in insulin resistance or central inflammation (Erion et al., 2014; Lambert et al., 2015; Miller & Spencer, 2014; Willette & Kapogiannis, 2014) causing impaired cerebrovascular reactivity (Miller & Spencer, 2014; Willette & Kapogiannis, 2014) or reduced synaptic plasticity (Miller & Spencer, 2014; Willette & Kapogiannis, 2014), respectively.

Erion, J. R., Wosiski-Kuhn, M., Dey, A., Hao, S., Davis, C. L., Pollock, N. K., & Stranahan, A. M. (2014). Obesity elicits interleukin 1-mediated deficits in hippocampal synaptic plasticity. Journal of Neuroscience, 34(7), 2618-2631.

Lambert, E. A., Straznicky, N. E., Dixon, J. B., & Lambert, G. W. (2015). Should the sympathetic nervous system be a target to improve cardiometabolic risk in obesity?

Miller, A. A., & Spencer, S. J. (2014). Obesity and neuroinflammation: A pathway to cognitive impairment. Brain, Behavior, and Immunity, 42, 10-21.

Willette, A. A., & Kapogiannis, D. (2014). Does the brain shrink as the waist expands? Ageing Research Reviews, 20, 86-97.

Experimental design

no comments

Validity of the findings

Method-

1. Are participants excluded differed from those included in terms of obesity, executive function, or episodic memory? Further, did the authors perform a sensitivity analysis to see whether the exclusion of participants modulates the association between obesity and executive function/episodic memory?

2. BMI is not a continuous variable. The reviewer suggests the authors transform individuals’ BMI into dummy variable (i.e., obese, normal-weight, lean) before entering them into regression analysis.

3. What are the performance indices for cognitive ability? Are performance based on response accuracy, points gained, reaction times, or items recalled? These information should not be omitted.

4. With regard to measures for obesity, have the authors considered %body fat or adiposity fat? These two measures has been adopted in relevant studies in this area (Kamijo, Khan, et al., 2012; Kamijo, Pontifex, et al., 2012).

Kamijo, K., Khan, N. A., Pontifex, M. B., Scudder, M. R., Drollette, E. S., Raine, L. B., ... & Hillman, C. H. (2012). The relation of adiposity to cognitive control and scholastic achievement in preadolescent children. Obesity, 20(12), 2406-2411.

Kamijo, K., Pontifex, M. B., Khan, N. A., Raine, L. B., Scudder, M. R., Drollette, E. S., ... & Hillman, C. H. (2012). The association of childhood obesity to neuroelectric indices of inhibition. Psychophysiology, 49(10), 1361-1371.

5. There are a number of background variables which should also be considered in your study, such as diet or physical activity. Diet has been suggested a key factor in the vicious cycle model (Kanoski & Davidson, 2011; Yeomans, 2017), while physical activity may counteract the detrimental effect of obesity on cognition (Kim & Park, 2018).

Kanoski, S. E., & Davidson, T. L. (2011). Western diet consumption and cognitive impairment: Links to hippocampal dysfunction and obesity. Physiology & Behavior, 103(1), 59-68.

Yeomans, M. R. (2017). Adverse effects of consuming high fat–sugar diets on cognition: Implications for understanding obesity. Proceedings of the Nutrition Society, 76(4), 455-465.

Kim, T. W., & Park, H. S. (2018). Physical exercise improves cognitive function by enhancing hippocampal neurogenesis and inhibiting apoptosis in male offspring born to obese mother. Behavioural Brain Research, 347, 360-367.

6. Rather than entering all background variables into the regression analysis. The reviewer suggests the authors perform bivariate correlations first to see which background variables are correlated with the predictor and/or dependent variables, and then enter these variables into regression analysis. The reviewer believes that this way of analysis is more logical.

7. Please justify the rationale of regression analysis. It is necessary to clarify why demographics were entered into the first model, why comorbidities of obesity were entered into the second, and so on....
Likewise, please justify why you are interested in the interaction between obesity and age.

Additional comments

The current manuscript has addressed an interesting and important topic. Key findings are that waist-to-hip ratio, but not BMI, is a valid predictor of deficits in executive function and episodic memory across the adulthood after adjusting for the influence of demographics, comorbid health issues, health behaviors, personality traits, and self-perceived obesity. Novelty of the present study is the comparison between BMI and WHR as an appropriate measure of obesity in investigating the relation with executive function and episodic memory. Overall, the manuscript is well-written. The authors should be credited by their efforts in conducting study with such large sample. Reservation, however, should be given to the validity of findings. My concerns lay on the authors’ decisions in 1) selecting indices of obesity, 2) appropriateness and rationale of data analysis, and 3) controlling of potential confounders. All these issues should be answered (or justified) before the merit of this paper could be revealed and contribute to the literature.

Reviewer 3 ·

Basic reporting

no comment

Experimental design

no comment

Validity of the findings

no comment

Additional comments

This manuscript is clearly written and well organized. The results were appropriately interpreted and discussed. The authors, however, need to add information about the software they used for statistical analysis in the text.

---

## Round 0.2 · Minor Revisions

I have now received two reviewers’ comment and both reviewers were generally satisfied with your reply and revisions from previous comments. However, a few minor issues remained to be addressed before I can accept your manuscript. Please take care of these issues and provide a point by point reply in addition to the revised manuscript.

Tsung-Min Hung, PhD., FNAK
PeerJ editor
Research chair professor,
Department of Physical Education,
National Taiwan Normal University

Reviewer 1 ·

Basic reporting

No comment.

Experimental design

No comment.

Validity of the findings

No comment.

Additional comments

The authors did a great job revising the paper. All of my concerns have been alleviated with this revision. The biggest shortcoming of the study i.e., BMI calculated by participants’ self-reported weight and height, has been mollified by additional data. Thus, I would recommend for publishing this manuscript.

I have some minor comments listed below.

1. L344: Please change age into sex.
2. Table 1: Please add a unit of household income.

·

Basic reporting

1. Prior comment 3, Table 3 & 4- Tables should be self-explanatory. The reviewer suggests the authors to explicitly elaborate what variables were included into model 1-5 in both tables. This should make readers catch up the tables more quickly.

2. Prior comment 4- the reviewer appreciates the summary table of cognitive assessments and performance indices; yet, the reviewer also suggests the authors to provide a table summarizing descriptive stats of performance in each cognitive tasks for clarity.

3. Prior comment 10- Please specify how physical activity was measured? By subjective or objective measurement tool?

Experimental design

No comment

Validity of the findings

No comment

Additional comments

The authors have done a great job in revising the manuscript! The reviewer has only several minor concerns before the manuscript is ready for publication.

---

## Round 0.3 · accepted · Accept

I have read through your reply to the reviewer's comment and your revised manuscript. I am satisfied with your response and decided that there is no need to send to the reviewer. You and your coauthors have my congratulations. Thank you for choosing PeerJ as a venue for publishing your research work and I look forward to receiving more of your work in the future.

Tsung-Min Hung, PhD., FNAK
PeerJ editor
Research chair professor,
Department of Physical Education,
National Taiwan Normal University

#